# Transcriptome Analysis of *Sogatella furcifera* (Homoptera: Delphacidae) in Response to Sulfoxaflor and Functional Verification of Resistance-Related P450 Genes

**DOI:** 10.3390/ijms20184573

**Published:** 2019-09-15

**Authors:** Xue-Gui Wang, Yan-Wei Ruan, Chang-Wei Gong, Xin Xiang, Xiang Xu, Yu-Ming Zhang, Li-Tao Shen

**Affiliations:** 1Biorational Pesticide Research Lab, Sichuan Agricultural University, Chengdu 611130, China; 2Sichuan Provincial Plant Protection Station, Department of Agriculture, Chengdu 610041, China

**Keywords:** *Sogatella furcifera*, sulfoxaflor, transcriptome, cytochrome P450 monooxygenase, RNA interference

## Abstract

The white-back planthopper (WBPH), *Sogatella furcifera*, is a major rice pest in China and in some other rice-growing countries of Asia. The extensive use of pesticides has resulted in severe resistance of *S. furcifera* to variety of chemical insecticides. Sulfoxaflor is a new diamide insecticide that acts on nicotinic acetylcholine receptors (nAChRs) in insects. The aim of this study was to explore the key genes related to the development of resistance to sulfoxaflor in *S. furcifera* and to verify their functions. Transcriptomes were compared between white-back planthoppers from a susceptible laboratory strain (Sus-Lab) and Sus-Lab screened with the sublethal LC_25_ dose of sulfoxaflor for six generations (SF-Sel). Two P450 genes (*CYP6FD1* and *CYP4FD2*) and three transcription factors (*NlE78sf*, *C2H2ZF1* and *C2H2ZF3*) with upregulated expression verified by qRT-PCR were detected in the Sus-Lab and SF-Sel strains. The functions of *CYP6FD1* and *CYP4FD2* were analyzed by RNA interference, and the relative normalized expressions of *CYP6FD1* and *CYP4FD2* in the SF-Sel population were lower than under dsGFP treatment after dsRNA injection. Moreover, the mortality rates of SF-Sel population treated with the LC_50_ concentration of sulfoxaflor after the injecting of dsRNA targeting *CYP6FD1* and *CYP4FD2* were significantly higher than in the dsGFP group from 72 h to 96 h (*p* < 0.05), and mortality in the *CYP6FD1* knockdown group was clearly higher than that of the *CYP4FD2* knockdown group. The interaction between the tertiary structures of *CYP6FD1* and *CYP4FD2* and sulfoxaflor was also predicted, and *CYP6FD1* showed a stronger metabolic ability to process sulfoxaflor. Therefore, overexpression of *CYP6FD1* and *CYP4FD2* may be one of the primary factors in the development of sulfoxaflor resistance in *S. furcifera*.

## 1. Introduction

The white-back planthopper (WBPH), *Sogatella furcifera* (Horváth) (Homoptera: Delphacidae), is an important insect pest in rice-growing countries in Asia [1,2] that seriously affects rice yields by sucking juice from the rice stem, resulting in slow growth, yellowing, and even lodging and a phenomenon known as‘hopper burn’, leading to the death of rice plants in severe cases and crop failure. In addition, recent studies have shown that the southern rice black-streaked dwarf virus (SRBSDV) can be transmitted in the main rice-growing areas of Asia, including China, Vietnam, and Japan, through the stylets of *S. furcifera* when injecting juice containing the SRBSDV into the stems of healthy rice, causing great rice yield losses. *S. furcifera* is a typical *r*-strategy pest and can reproduce rapidly when conditions are suitable [3,4,5]. It can also migrate long distances to adapt to environmental changes [6]. Over the past few decades, chemical pesticides have been the main strategy for controlling this pest [7]. The long-term use of insecticides against the rice planthopper has resulted in the development of resistance and population increases [8]. *S. furcifera* has now developed different levels of resistance to organophosphorus, carbamate, phenylpyrazole, neonicotinoid, pyrethroid, and insect growth regulator insecticides [9,10,11,12,13]. When insects are successively exposed chemicals, insecticide resistance is a natural adaptability feature and one of the most important factors is the enhanced detoxification metabolic ability by enzymes to insecticides, such as mixed-function oxidase (MFO), carboxylesterase (CarEs) and glutathione *S*-transferase (GSTs) [14], especially those for cytochrome P450 monooxygenases (P450s) activity, could play a major role in the detoxification of insecticides in a number of insect pests [15,16].

Sulfoxaflor is an insecticide produced by Dow AgroSciences (DAS) from a new chemical class of sulfoximines, which act on nicotinic acetylcholine receptors (nAChRs) in the insect nervous system [17,18], and is the first commercial agrochemical to be used for the control of a broad range of sap-feeding insect pests [19]. Although several other chemically distinct classes of insecticides (spinosyns, neonicotinoids, nereistoxin analogs) also act on nAChRs, sulfoxaflor presents some special biological characteristics, such as an absence of cross-resistance between sulfoxaflor and the other nAChR-acting insecticides, which make it highly effective against a wide range of sap-feeding insects, especially against aphids such as the green peach aphid [20]. Sulfoxaflor is also effective against insect pests that are resistant to other classes of insecticides, including many insects that are resistant to neonicotinoids [21]. In 2012, sulfoxaflor was first registered for the control of Miridae pests in cotton cultivation in Arkansas, Louisiana, and Mississippi in the United States, and in 2013, Dow AgroSciences introduced sulfoxaflor to China for the control of cotton aphids, wheat aphids, scale insect pests in citrus, and Delphacidae insect pests in rice [22]. As a relatively recently developed insecticide, there is still a risk that the efficiency of sulfoxaflor may become compromised through various mechanisms, e.g., through modification of the target site of neonicotinoid insecticides or positive sublethal effects (hormesis) in exposed individuals [23]. Thus, it is crucial to understand the resistance mechanisms of insects related to this insecticide before its widespread use in integrated pest management (IPM).

In this study, we examined toxicity and transcript profiles in a susceptible laboratory (Sus-Lab) strain of *S. furcifera* and the Sus-Lab strain in which resistance was continuously induced by treatment with the sublethal LC_25_ dose of sulfoxaflor for six generations (SF-Sel). We analyzed the genes showing upregulated expression and verified the results using qRT-PCR. Furthermore, the functions of two P450 genes, *CYP6FD1* and *CYP4FD2*, were also analyzed with RNA interference technology through the design of suitable dsRNAs, whose silencing specificity was confirmed using qRT-PCR, and mortality was determined in nymphs treated with the dsRNAs combined with sulfoxaflor. The objectives of this study were to explore the risk of the development of resistance to sulfoxaflor, to conduct a preliminary investigation for the functional verification of P450 genes induced by sulfoxaflor, and to manage the development of resistance through the design of suitable dsRNAs for the silencing of significantly upregulated genes in the future.

## 2. Results

### 2.1. Toxicity of Sulfoxaflor in S. furcifera

The LC_25_ for the Sus-Lab strain was estimated at 2.102 µg/mL, and a 2.06-fold decrease in the susceptibility level was observed in the SF-Sel strain, which presented LC_50_ value of 7.284 µg/mL for sulfoxaflor, compared with the LC_50_ value of 3.544 µg/mL for Sus-Lab.

### 2.2. Synergism Experiment

According to the results of the synergism experiment, when the Sus-Lab strain was treated with the synergistic agents triphenyl phosphate (TPP) acting on CarEs, Diethyl maleate (DEM) on GSTs and piperonyl butoxide (PBO) on cytochrome P450 monooxygenases (P450s), all synergistic treatments showed strong synergistic effects compared with treatment without a synergistic agent, and their toxicities were clearly enhanced (not overlapping the 95% confidence interval, 95%CI), with synergism ratio (SR) values of 1.495, 1.205 and 1.211, respectively, while as the toxicity showed no difference between the synergistic treatments (overlapping 95% CI). The toxicity of the SF-Sel strain treated with the synergistic agent PBO was significantly enlarged compared with those in the other two treatments involving TPP and DEM (not overlapping 95% CIs for the other two synergistic agents). The SR value reached 5.328 and was significantly higher than those in the treatments involving TPP and DEM, for which SR values were 2.689 and 2.283, respectively (Table 1).

### 2.3. Detoxification Enzyme Activity

To further determine the potential role of detoxification enzymes in the development of resistance of *S. furcifera* to sulfoxaflor, both the Sus-Lab strain, either untreated or treated with a synergistic agent (DEM, TPP or PBO), and the SF-Sel strain, either untreated or treated with a synergistic agent (DEM, TPP or PBO), were analyzed to determine the activities of CarEs, GSTs and P450s. As shown in Figure 1, the activity of CarEs in the SF-Sel strain treated with TPP was the highest (0.9616 mmol × min^−1^× mg pro^−1^), followed by those in the SF-Sel strain (0.8257 mmol × min^−1^× mg pro^−1^), the Sus-Lab strain treated with TPP (0.8187 mmol × min^−1^× mg pro^−1^) and the Sus-Lab strain (0.7397 mmol × min^−1^× mg pro^−1^). However, there was no significant difference in CarE activity under any of the treatments (*p* > 0.05). The GSTs activity of the SF-Sel strain was the greatest, at 1.0103 mmol × min^−1^× mg pro^−1^, which was significantly different from the other three treatments (*p* < 0.05), where the activities ranged from 0.6824 to 0.8113 mmol × min^−1^× mg pro^−1^ and did not significantly differ from each other (*p* > 0.05). P450s activity was also the greatest in the SF-Sel strain (13.5345 nmol × min^−1^× mg pro^−1^, which was significantly different from those under the other treatments (*p* < 0.05)), followed by the SF-Sel strain treated with PBO (10.7384 nmol × min^−1^× mg pro^−1^). The P450s activities of the Sus-Lab strain and Sus-Lab strain treated with PBO were the weakest (5.6181 and 4.8470 nmol × min^−1^× mg pro^−1^), and both of these values were significantly different from those under the other treatments (*p* < 0.05) (Figure 1).

### 2.4. Illumina Sequencing and Read Assembly

The Sus-Lab and SF-Sel strains were each assessed in triplicate. The total numbers of reads (150 bp) obtained were 335,119,142 in the six samples, with over 50,653,904 reads for each sample. The proportion of reads containing duplicate sequences was 0.37% ~ 0.42%, and the proportion of low-quality reads was 1.23%~1.35%, including reads with > 10% Ns and abase number of Q ≤ 10 in > 50% of the total reads. After filtering out the linker sequences or low-quality reads, 329,449, and 482 clean reads were obtained, and the Q20 and Q30 base percentages of clean reads of were over 98.69% and 96.04%, respectively (Appendix A).

### 2.5. Transcriptome Data Splicing

The clean reads from the transcriptomes of the Sus-Lab and SF-Sel strains were composed of the mixed pools and were assembled into approximately 74,119 unigenes, with a longest unigene of 34,340 nt and a shortest unigene of 201 nt (Appendix A). The N50 value, at which the cumulative fragment length reaches 50% of the total fragment length, was 2043 nt, and there were 5552 unigenes of over 3000 nt. When the reads were compared with the unigenes, 3971 of the unigenes were found to exhibit more than 10,000 reads, while 34, 847 of unigenes presented only 11~100 reads (Appendix A).

### 2.6. Transcriptome Annotation

From to the 74,119 assembled unigenes, 24,719 of which were successfully annotated in the Nr database, and the species with the greatest number of homologous sequences included *Zootermopsis nevadensis* (2001 genes), *Bemisia tabaci* (1620 genes), *Cimex lectularius* (1500 genes), and *Halyomorpha halys* (1424 genes) (Appendix A). Additionally, 19,050 unigenes were annotated with the Swissprot database, 17,119 unigenes with the KOG database, and 11,788 unigenes with the KEGG database, and 10,516 unigenes were annotated in all four databases (Appendix A), among the 17,119 unigenes annotated with the KOG database, 6671 unigenes were classified as general function prediction only, accounting for the largest proportion, and 4867 unigenes were classified as being associated with signal transduction mechanisms (Appendix A).

### 2.7. Analysis of Gene Expression

The results indicated that the correlations of gene expression levels in the three samples (M1, M2 and M3) of the Sus-Lab strain (with correlation index of 0.99 to 1.00) were significantly higher than those in the SF-Sel strain (S1, S2 and S3 samples, with a correlation index of 0.85 to 0.94) (Figure 2A). Principal component analysis (PCA) of the six samples also showed that there was a significant clustering relationship on PC1 between the gene expression levels in the samples of the susceptible strains and the SF-Sel strain. Additionally, the degree of the contribution of PC1 (88.7%) was significantly higher than that of PC2 (7.9%) (Figure 2B).

### 2.8. Cluster Analysis of DEGs

Based on the screening of DEGs with under criteria of an FDR < 0.05 and log2FC| > 1, 786 DEGs were screened from the SF-Sel strain, 557 of which were upregulated, while 229 were downregulated compared with those in the Lab-Sus strain (Figure 3A,B).

#### 2.8.1. GO Enrich

The up- or downregulated unigenes were enriched and assessed in the three Gene Ontology (GO) categories of biological process, cellular component and molecular function. The enrichment of upregulated unigenes from the SF-Sel strain was found in the Sus-Lab strain. Additionally, the number of unigenes of upregulated unigenes of the binding class in the molecular function category was highest, at 16, and the P450 genes were associated iron ion binding; with up- and downregulated P450 gene were enriched in the binging class (Appendix A).

#### 2.8.2. Enrichment of DEGs in the KEGG Database

A total of 84 differentially expressed unigenes were enriched according to the KEGG database, which were related to organismal systems, genetic information processing, and metabolism, etc. The enriched DEGs found in the KEGG database were mainly included in categories such as “microbial metabolism in diverse environments (with a *p*-value of 0.00337)”, “cardiac muscle contraction (*p*-value of 0.00153), “oxidative phosphorylation (*p*-value of 0.00114), “metabolic pathways (*p*-value of 0.000322)”, “ribosome (*p*-value of 0.0000753), and “phototransduction-fly *(p*-value of 7.27 × 10^−8^). Additionally, the greatest number of unigenes enriched in metabolic pathways was 44 (Figure 4).

### 2.9. Screening the Candidate Genes

The results showed that among the 74,119 unigenes, there were 198 unigenes involved in the detoxification and metabolism of foreign substances, and 138 were labeled cytochrome P450s, enriching the detoxification and metabolism genes of the white-backed planthopper. Compared with the Sus-Lab strain, the SF-Sel strain exhibited 7 significantly upregulated unigenes annotated as P450 genes, and a total of 4 gene types were found by NCBI alignment. One of the significantly upregulated unigenes was annotated as anorganic cation transporter, and 4 of the significantly upregulated unigenes were annotated as transcription factors (Table 2).

### 2.10. P450 Diversity Analysis

Ten motifs (motif 1~motif 10) composed of very conservative amino acid residues were found in 54 of P450 genes (22 from our transcriptome data and other 32 downloaded from NCBI) of white-backed planthopper by meme search (http://meme-suite.org/tools/meme), on the contrary 24 motifs annotated with different functions were achieved by motif search (https://www.genome.jp/tools/motif/) (e.g., P450). Not only highly homologous sequences but also functional domains such as FAD_binding_1, Flavodoxin_1 and NAD_binding_1 domains were found in AHM93009.1 (the Sequences of NADPH-cytochrome P450 reductase in *S. furcifera* downloaded from NCBI) and *unigene0040669* (Figure 5). The results indicated that these sequences were similar in function, and GO annotation showed that both were classified as NADPH-reductases. The other genes all exhibited P450 functional domains and the conserved structural domains of motif1 and motif3. Motif3 presented absolutely conserved EXXR residues, and their general topography and structural folding were highly conserved. The heme-binding loop (with an absolutely conserved cysteine that serves as the 5th ligand for the conserved heme iron core) was composed of a coil known as the ‘meander’, a four-helix bundle, helices J and K, and two sets of beta-sheets. Additionally, the conserved structural domains of motif7, motif4, motif8, motif6, motif5, motif3, motif1, and motif2 were found in *CYP6FD1* and *CYP4FD2*, but the conserved structural domains of motif9 and AAA (ATPases superfamily, consisted of ATP binding site, Walker A and B) were only found in *CYP4FD2* and *CYP6FD1*, respectively (Appendix A).

### 2.11. Quantitative PCR (qRT-PCR)

The results indicated that the relative normalized expressions of *CYP6FD1* and *CYP4FD2* in the SF-Sel strain were 5.22- and 2.99-fold higher, respectively, than in the Sus-Lab strain, which were significantly higher than those of *CYP4FD1* and *CYP6FD2* (*p* < 0.05). The relative normalized expression of the *NlE78sf* transcription factor was highly significantly increased by 1.91-fold (*p* < 0.01), while the transcription factors *C2H2ZF1* and *C2H2ZF3* showed 1.53-and 1.30-fold increases, respectively, compared with the Sus-Lab strain (*p* < 0.05) (Figure 6).

### 2.12. Functional Analysis of CYP6FD1 and CYP4FD2 via RNAi

The results indicated that the relative normalized expression of *CYP6FD1* and *CYP4FD2* was significantly lower than that of dsGFP (*p* < 0.05) at 24 h after treatment, only reaching 0.72- and 0.68-fold, respectively. With prolongation of the interference time (48–72 h), the RNAi efficacy of *CYP6FD1* and *CYP4FD2* dsRNA became greater, with relative normalized expression of 0.42- and 0.55-fold for *CYP6FD1* and *CYP4FD2,* respectively, at 48 h to 72 h after injection, while that of *dsCYP6FD1* was higher than that of *dsCYP4FD2* (*p* < 0.05). However, the RNAi efficacy of *CYP6FD1* and *CYP4FD2* dsRNA gradually decreased, with the relative normalized expression of 0.61- and 0.71-fold for *CYP6FD1* and *CYP4FD2*, respectively, at 96 h after injection (Figure 7A).

The results of the bioassay showed that after 72 h of treatment with sulfoxaflor at the LC_50_, the mortality of the SF-Sel strain treated with *CYP6FD1* dsRNA was 65.89% ± 5.38, which was significantly higher than the mortality of the same strain treated with *CYP4FD2* dsRNA (48.14% ± 4.33) and that in the control treatment (dsGFP, 38.95% ± 3.07) (*p* < 0.05). With prolongation of the interference time (96 h), the mortality of the *CYP6FD1* and *CYP4FD2* dsRNA treatment groups increased, reaching 62.01% ± 2.12~77.94% ± 5.30, respectively; this difference was also significant (*p* < 0.05), and mortality in these groups was significantly higher than under treatment with dsGFP (48.14% ± 0.93) (Figure 7B).

### 2.13. Interaction of the Tertiary Structure of CYP6FD1 and CYP4FD2 with Sulfoxaflor

The *CYP6FD1* domain included these amino acids ILE97, PHE105-GLY109, TYR111, and HIS122-SER126, and the absolute conserved residues of GLU444 and ARG447 were located near the entrance of the active pocket and the N-segment where heme binding occur. The totals core between *CYP6FD1* and sulfoxaflor was 5.8954 (crash of −0.7686 and polar of 0.0277). However, the domain of *CYP4FD2* was mainly composed of the LEU103, LYS106-LYS108, ALA110-LYS112, and LEU123 amino acids, and the key amino acids PHE317 and ILE459 bind with sulfoxaflor through noncovalent bonds, with a total score of 5.2800 (crash of −1.4457 and polar of 0.5769). Moreover, the absolutely conserved residues of GLU75 and ARG78 were distant from the active pockets and the heme binding region (Figure 8).

## 3. Discussion

Although rice planthoppers in China are still sensitive to fluorodinitrile, our results showed that the toxicity of sulfoxaflor to *S. furcifera* decreased when the Sus-Lab strain was successively screened with the sublethal dose of sulfoxaflor, which indicated that *S. furcifera* presented some resistance to sulfoxaflor. Liao et al. [24] monitored the resistance levels of *Nilaparvata lugens* to sulfoxaflor from 2013 to 2016 in China and found that all field-collected populations were still sensitive, with LC_50_ values ranging from 1.63 to 13.20 mg/L (resistance ratio from 0.8 to 6.8-fold). Liao et al. [25] continuously screened *N. lugens* with a sublethal dose of sulfoxaflor in approximately 39 intervals and finally obtained an extremely sulfoxaflor-resistant strain with a resistance ratio of 183.6-fold. Ma et al. [26] performed continuous selection of *Aphis gossypii* gradually increasing LC_50_ concentrations of sulfoxaflor based on bioassays of the parental generations for a total of 27 generations in the laboratory, finally resulting in a 366.4-fold resistance ratio compared with the susceptible strain. Therefore, there is an extremely high risk of insects developing resistance to sulfoxaflor, and it is necessary to perform resistance monitoring in field populations, along with the investigation of resistance mechanisms and cross-resistance to design integrated pest management strategies.

Insecticide resistance is inevitable after the application of insecticides, and the main reasons include a reduced penetration rate, increased detoxification and metabolism of insecticides (MFOs [27], CarEs [28], GSTs [29]) and decreased sensitivity at the target site. Wei et al. [30] found that PBO and TPP could increase bifenthrin toxicity in resistant *A. gossypii* Glover strains by 2.38- and 4.55-fold, respectively. Liao et al. [25] showed that the toxicity of sulfoxaflor to sulfoxaflor- resistant *N. lugens* (Stål) showed a synergistic effect with PBO resulting in a 2.69-fold relative synergistic ratio, and the P450 enzyme activity of SFX-SEL was increased 3.50-fold compared with that in the unselected strain (UNSEL). Mao et al. [31] also reported that a resistant strain (NR) with a high nitenpyram resistance level (164.18-fold) and cross-resistance to sulfoxaflor (47.24-fold) showed a 3.21-fold increase in P450 activity compared to that in NS, and resistance also showed a synergistic effect (4.03-fold) with the inhibitor PBO, suggesting a role of P450. Our results further demonstrated that the inhibitors PBO, TPP, and DEM showed some synergism with sulfoxaflor regarding the toxicity and inhibition of the activities of three types of metabolic detoxification enzymes in the Sus-Lab and SF-Sel strains; this effect was especially strong for the inhibitor PBO in the SF-Sel strain.

Normally, the contribution of the overexpression of detoxification metabolism genes to an increased detoxification ability, especially which of the P450 genes related to insecticide detoxification metabolism, is the main reason for insect resistance to insecticides [32,33,34]. Jones et al. [35] reported that the resistance of the ALM07 strain of B-biotype populations of *Bemisia tabaci* adults to imidacloprid reached 180-fold, and the relative normalized expression of the resistance gene *CYP6CM1* in adults and nymphs reached 4.2- and 200-fold in the resistant strain, respectively. The overexpression of *CYP6AY1* contributes to the development of resistance to imidacloprid in *N. lugens* [36]. Liao et al. [25] and Mao et al. [31] also demonstrated that the reducing expression of *CYP6ER1* in sulfoxaflor-resistant strain through RNAi could significantly increase its’ susceptibility to sulfoxaflor. Our transcriptome data and qRT-PCR results also indicated that two P450 genes, *CYP6FD1* and *CYP4FD2,* and three transcription factors, *NlE78sf*, *C2H2ZF1* and *C2H2ZF3*, were clearly upregulated in the SF-Sel strain. The RNAi results also showed that when 3rd-instar nymphs were injected with the *CYP6FD1* and *CYP4FD2* dsRNA, the relative expression of *CYP6FD1* and *CYP4FD2* was decreased, causing the insects to be more sensitive and ultimately to show higher mortality compared with negative dsGFP control treatment. However, it is still uncertain which transcription factors are mainly responsible for regulating the overexpression of *CYP6FD1* and *dsCYP4FD2*, and require further study in the future.

The P450s area multi-enzyme complex, and the first step in the metabolism of exogenous toxic substances is recognition by a CYP protein, which binds the toxin; then, electrons are transferred by electron donors to exogenous REDOX substances [37]. At present, the examination of P450 structure in insects generally concentrates on assessing highly conserved sequence motifs, such as the residue pairs WxxxR in helix C, CxxT in helix I, ExxR in helix K, RxxF in the meander region, and FxxGxRxCxG in the canonical heme-binding domain [38]. Our research showed that the molecular structure of sulfoxaflor was surrounded by the active pocket of *CYP6FD1*, while the active pocket was located near the heme-binding region. This protein exhibits a predicted active site structure with an oval shape [39], a large volume, and large substrate channels, allowing sulfoxaflor to fit the active site cavity. The spacious cavity of P450 enzymes enables larger molecules to access the heme-bound oxygen of the reaction center; therefore, we hypothesize that *CYP6FD1* could present a greater metabolic ability than *CYP4FD2* [40,41].

On the basis of our results, the main findings show that it is likely that *S. furcifera* will develop resistance to sulfoxaflor and that upregulation of detoxification enzymes such as P450s is a likely mechanism. However, the authors also show that the toxicity of sulfoxaflor is increased by using synergistic agents, so perhaps this is one possible approach that could be used in the field to prevent rapid development of resistance to this compound. Meanwhile, we also find that two main P450 genes (*CYP6FD1* and *CYP4FD2*) could be related to the development of resistance of *S. furcifera* to sulfoxaflor. Our results should provide a foundation for subsequent efforts to investigate the expression of *CYP6FD1* and *CYP4FD2* in heterologous expression systems, such as baculovirus- infected Sf9 cells, and metabolic processes in vitro and transcriptional regulation of the two genes in the further investigations.

## 4. Materials and Methods

### 4.1. Insects and Insecticide

The susceptible laboratory (Sus-Lab) strain of WBPH (*S. furcifera*) established in our laboratory was obtained from the research group of Prof. Li, College of Plant Protection of Hunan Agricultural University (Changsha, China) in 2016, where the strain had been reared in the laboratory without exposure to any insecticide since 2009. All stages were maintained on rice seedlings under standard conditions of a temperature of 27 ± 1 °C, relative humidity (RH) of 70–80% and a light/dark cycle of 16:8 h. Sulfoxaflor (95%, technical grade) was purchased from Dow AgroSciences (Shanghai) Co., Ltd. China (Shanghai, China).

### 4.2. Selection of the SF-Sel strain with a Sublethal Dose of Sulfoxaflor

The toxicity of sulfoxaflor to *S. furcifera* was performed using the rice seedling dipping method, with some modifications [42]. First, technical grade sulfoxaflor was dissolved in acetone, and a series of suitable concentrations (i.e., 1, 2, 4, 6, and 8 µg/mL) were prepared with 0.1% Triton X-100; the 0.1% Triton X-100 solution alone was used as the blank control. Four to five leaves of rice seedlings were cleaned with water and air-dried at room temperature. Fifteen rice seedlings were bundled together, immersed in the diluted solution for approximately 30 s, and then dried at room temperature. Second, moistened cotton was wrapped around the rice roots, which were immobilized in a 500 mL plastic cup. Then, fifteen 3rd-instar nymphs were transferred to each plastic cup, and all treatments were set up in triplicate. All treatments were performed under standard environmental conditions (26 ± 1 °C, 85 ± 10% R.H., 14:10 L: D), and mortality was recorded after 96 h of treatment. Individual nymphs were considered dead if they did not show movement after being slightly nudged with a #26 soft brush. Probit analyses were conducted using a Statistical Analysis System (SAS) software to calculate the slope, LC50, 95% CI, and χ2 values of sulfoxaflor or sulfoxaflor plus synergistic agents after 96 h of treatment [16]. Then, continuous selection with the sublethal LC_25_ dose of sulfoxaflor was performed for six generations in the SF-Sel strain.

### 4.3. Test for Synergism

The synergism bioassays for the Sus-Lab and SF-Sel strains of *S. furcifera* to sulfoxaflor were performed as described by Mu et al. [13] with some modifications. Three synergistic agents, DEM, TPP and PBO, were dissolved with acetone and diluted with Triton X-100 to the highest possible concentrations showing no adverse effect on the tested insects (PBO, 30 µg/mL; TPP, 160 µg/mL; DEM, at 300 µg/mL), after which rice seedlings into the synergistic treatment solutions for 30 s and naturally dried them. Then, approximately 300 of 3rd-instar nymphs were transferred to the rice seedlings treated with each synergistic agent for approximately 2 h. The remaining procedures were similar to the rice seedling dipping method as described above.

### 4.4. Enzyme Assays

To evaluate the potential role of the detoxification enzymes of *S. furcifera* in resistance to sulfoxaflor, the activities of CarEs, GSTs and P450s in the 3rd-instar nymphs of the Sus-Lab and SF-Sel strains treated with synergistic agents (TPP or DEM or PBO) were determined.

CarE activity was determined according to the method described by van Asperen [43]. Twenty 3rd-instar nymphs were placed in a centrifugal tube and stored in liquid nitrogen as quickly as possible, then homogenized on ice in 2 mL of homogenization buffer (0.04 mol/L phosphate buffer, pH 7.0) using a 5 mL glass homogenizer and centrifuged at 4 °C, 10,000× *g* for 15 min using a 5417R centrifuge (Eppendorf, Germany). The supernatant was subsequently transferred to a clean Eppendorf tube as the crude enzyme solution. A mixture of 0.45 mL of phosphate buffer (0.04 mol/L, pH 7.0), 1.8 mL of 3 × 10^−4^ mol/L α-NA solution (containing 3 × 10^−4^ mol/L physostigmine) and 50 µL of diluted enzyme liquid was added to each tube, followed by mixing and then incubation in a water bath at 30 °C for 15 min, after which the process was stopped with 0.9 mL of staining solution (0.2 g of fast blue-B salt in 20 mL of distilled water plus 50 mL of 5% sodium dodecyl sulfate). The absorbance values were recorded at 600 nm after 5 min in a UV 2000-Spectrophotometer (Unic [Shang Hai] Instruments Incorporated, Shanghai, China).

GST activity was determined using 1-chloro-2, 4-dinitrobenzene (CDNB) as a substrate according to the method of Wang et al. [16] with minor revisions. Twenty 3rd-instar nymphs were homogenized on ice in homogenization buffer (0.1 mol/L phosphate buffer containing 1.0 mmol/L EDTA, pH 6.5) and centrifuged at 10,800 rpm at 4 °C for 10 min, after which the supernatant was used as an enzyme source. A mixture including 2470 µL of phosphate buffer (0.1 mol/L, pH 6.5), 90 µL of CDNB (15 mmol/L), 50 µL of the enzyme source and 90 µL of reduced GSH (30 mmol/L) were added to a 5 mL centrifuge tube, which was promptly shaken. The OD value was recorded at 340 nm for 2 min and calculated as ∆A_340_/min.

P450 activity was assayed using the method of Rose et al. [44] with some modifications. One hundred and fifty 3rd-instar nymphs were homogenized on ice in 2 mL of homogenization buffer (0.1 mol/L, pH 7.6, containing 20% glycerol, 0.1 mmol/L EDTA, 0.1 mmol/L DTT, and 0.4 mmol/L PMSF). The homogenates were centrifuged at 4 °C at 10,000× *g* for 10 min using a 5417R centrifuge (Eppendorf, Germany) to obtain the supernatant, which was used as the crude enzyme. Then, 100 μL of 4-nitroanisole (2 × 10^−3^ mol/L) was added to the cell culture plate and mixed with 90 μL of crude enzyme liquid, followed by incubation for 3 min at 27 °C in a water bath kettle and the addition of 10 μL of NADPH (9.6 × 10^−3^ mol/L) for reaction. The changes in the OD value were recorded at 405 nm (Model 680 Microplate Reader, Bio-Rad) every 20 s for 2 min. A standard curve was generated using p-nitrophenol, and the specific activity of P450s was finally calculated as nanomoles of p-nitrophenolper minute per milligram of protein.

All treatments were set up with three samples (tubes) as biological repetitions, and each enzyme sample was individually prepared. Each assay of enzymatic activity was replicated three times as mechanical repetitions for each enzyme sample. The total protein content of the enzyme solution was determined by the Bradford method [45] using bovine albumin as a standard. The activities of CarEs, GSTs and P450s were analyzed using unpaired Student’s t-tests, and the significance level of the results was set at *p* < 0.05.

### 4.5. Transcriptome Analysis

#### 4.5.1. Library Construction and Sequencing, Illumina Read Processing, and Assembly and Annotation of Unigenes

According to the manufacturer’s protocol for the TRIzol^®^ Reagent (Invitrogen™, ThermoFisher Scientific, USA), approximately 100 nymphs and adults that had either been continuously selected with the LC_25_ dose of sulfoxaflor for six generations (SF-Sel) or not (Sus-Lab) were used for total RNA extraction. cDNA library construction and sequencing, Illumina read processing, assembly and bioinformatics analysis, and the annotation of unigenes, including protein functional annotation, pathway annotation, COG/KOG functional annotation and Gene Ontology (GO) annotation, etc., were performed as described by Wang et al. [46].

#### 4.5.2. Gene Expression and Differential Gene Enrichment

The expression of unigenes was calculated with the of RPKM (reads per kb per million reads) method [47] according to the following formula: RPKM(A) = (1000000 ∗ C)/(N ∗ L/1000)

The RPKM(A) value stands for the expression of gene A; C values stands for the number of reads that uniquely aligned to gene A; N values stands for the total number of reads that uniquely aligned to all genes, and L stands for the number of bases on gene A.

According to the gene expression represented by the RPKM values for each sample, the significant differentially expressed genes (DEGs) among the samples were screened with edge R. The screening criteria were an FDR < 0.05 (*p*-value after calibration by FDR) and |log2FC| > 1, and GO functional analysis and KEGG pathway analysis were performed based on the results for the DEGs.

#### 4.5.3. Diversity and Collinearity of *S. furcifera* P450 Genes

Twenty-two relatively complete P450 amino acid sequences obtained from the transcriptome were compared with thirty-two P450 amino acid sequences of the white-back planthopper downloaded from NCBI and analyzed for the conserved functional domains with the motif (https://www.genome.jp/tools/motif/) and meme (http://meme-suite.org/tools/meme) tools, and their phylogenetic tree was constructed by using MEGA 6.0 software with the default settings and the neighbour-joining method. The results were visualized with TBtools software.

### 4.6. Quantitative PCR (qRT-PCR)

Total RNA of the Sus-Lab and SF-Sel strains was extracted using TRIzol reagent (Invitrogen™, ThermoFisher Scientific, USA) according to the instructions of the manufacturer’s kit, and the reverse transcription reaction was performed with a cDNA Synthesis for qPCR (One-Step gDNA Removal) kit according to the instruction manual. The cDNA was kept at −20 °C for qRT-PCR.

The cDNAs of four P450 genes (*CYP6FD1*, *CYP6FD2*, *CYP4FD1* and *CYP4FD2*), one transporter (*Unigene0036498*), four transcription factors (*NlE78sf*, *C2H2ZF1*, *C2H2ZF3* and *C2H2ZF2*) and one reference gene (*RPL9*) [48] from the Sus-Lab and SF-Sel strains were amplified by PCR with twelve pairs of corresponding primers (Table 3). The qRT-PCR system and procedure were as described by Wang et al. [49]. All experimental results were analyzed in three independent replicates, and the treatment means and variances were analyzed via one-way ANOVA with PROC GLM of the SAS program. All means were compared by least squared difference (LSD) tests at a Type I error = 0.05.

The biological function of *CYP6FD1* and *CYP4FD2* was verified through RNA interference as described by Mao et al. [31] and Wang et al. [49], with some modifications. A 168 bp fragment of *CYP6FD1*, 450 bp of *CYP4FD2* cDNA and a 657 bp green fluorescent protein (gfp) fragment were amplified by PCR using corresponding primer pairs (with the T7 promoter appended). PCR was performed with the primers listed in Table 4. The PCR products were purified for use as templates for dsRNA synthesis using the T7 MEGAscript kit (ThermoFisher, USA) according to the manufacturer’s instructions. The dsRNA concentration was measured using a spectrophotometer (Nanodrop) after 1:10 dilution of the dsRNA product in water and adjustment of the ultimate concentration to 4 ng/μL for injection.

### 4.7. Function Analysis of CYP6FD1 and CYP4FD2 via RNAi

Third-instar nymphs were used for dsRNA injection experiments. First, the tested insects were anesthetized with CO_2_ for approximately 30 s, and each insect received 120 ng (approximately 30 μL) of the dsRNA for each target gene using an UMP3/Nanoliter2010 microinjection device (World Precision Instruments, Sarasota, Florida), with dsGFP used as a negative control, and 300 3rd-instar nymphs were prepared to check the RNAi efficiency and bioassay for each gene. The relative expression of *CYP6FD1* and *CYP4FD2* was detected at 24, 48, 72 and 96 h after injection. For insecticide bioassays after RNAi, thirty 3rd-instar nymphs were collected 24 h after injection and sixty 3rd-instar nymphs for each treatment were transferred to rice seedlings that had been treated with the LC_50_ of sulfoxaflor in solution. Mortality was calculated at 72 h and 96 h after insecticide treatment. Three biological replicates were performed.

### 4.8. Prediction the Interaction of Tertiary Structure of CYP6FD1 and CYP4FD2 with Sulfoxaflor

To obtain information about how sulfoxaflor affects P450s, molecular docking between sulfoxaflor and the active sites of the target P450 proteins was carried out using the Surflex-Dock program in SybylX-2.0 version (Tripos Inc.) as previously described [50]. Surflex-Dock scores (total scores) were expressed in kcal/mol units to represent binding affinities [51,52].

### 4.9. Data Analysis

The relative normalized expression of the upregulated P450 genes in the Sus-Lab and SF-Sel strains, the efficacy of *CYP6FD1* and *CYP4FD2* knockdown in 3rd-instar nymphs of the SF-Sel strain by RNAi, and the mortality of larvae injected with dsRNA with or without the LC_50_ concentration of sulfoxaflor were compared using analysis of variance (ANOVA) followed by Duncan’s test for multiple comparisons (*p* < 0.05) with the SPSS version 17.0 software package (IBM).

## Figures and Tables

**Figure 1 ijms-20-04573-f001:**
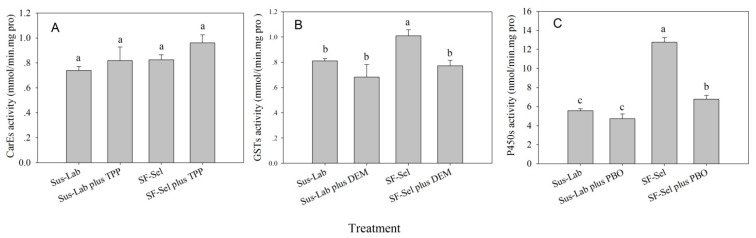
Synergistic effects of TPP, DEM and PBO on activity of detoxification enzymes: CarE (**A**), GST (**B**) and P450 (**C**) in 3rd-instar nymph of *S furcifera.* The activities of CarE, GST and P450 in 3rd-instar nymph of *S furcifera* are presented as the mean of three replications ± *SE*. Means followed by the same letters did not differ significantly (*p* > 0.05) according to the ANOVA test. The F_3, 8_ values of different treatments on CarE, GST and P450 in 3rd-instar nymph of *S furcifera* were 1.818, 5.410, 610.745, and the *p* values on CarE, GST and P450 in 3rd-instar nymph of *S furcifera* were = 0.222 > 0.05, = 0.025 < 0.05, = < 0.0001, respectively.

**Figure 2 ijms-20-04573-f002:**
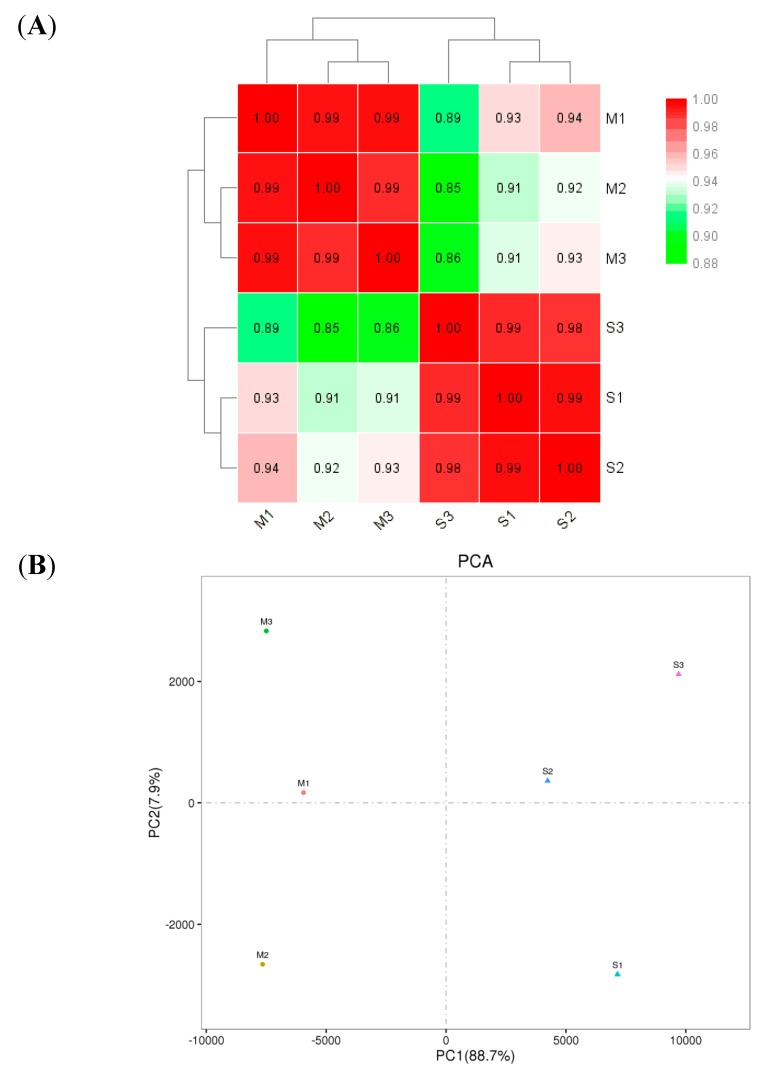
(**A**) Heat map of gene expression levels in the six samples. The darker the color is, the greater the correlation is. Three samples (M1, M2 and M3) are in the Sus-Lab strain, and other three samples (S1, S2 and S3) are in the SF-Sel strain. The same means as followed. (**B**) Principal component analysis (PCA) of six samples.

**Figure 3 ijms-20-04573-f003:**
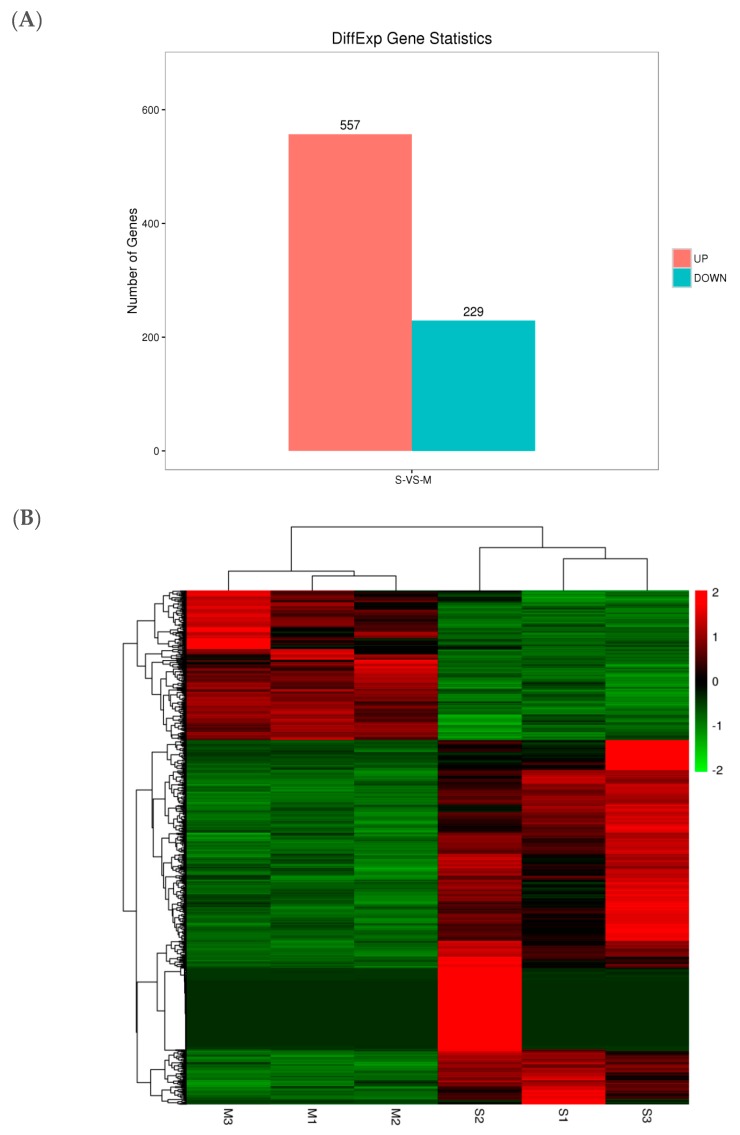
(**A**) The statistical maps of DEGs among S-vs-M. (**B**)The cluster heat maps of DEGs among S-vs-M. The column represents samples, and the row means genes; Color-scaled represents log 2 (fold change) values for resistant lines, the redder the color is, the higher the gene expression is, and on the contrary, the greener color is.

**Figure 4 ijms-20-04573-f004:**
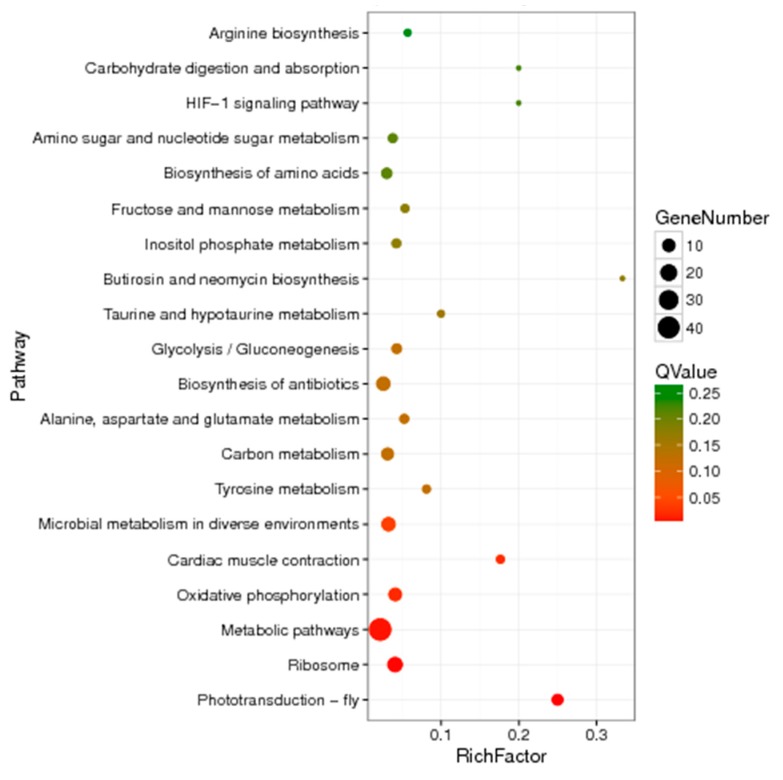
Bubble diagram of pathway enrichment of unigene on the different groups (Top 20). The abscissa means KEGG terms, and the ordinate means rich factor of each term. Red color means the terms of significant enrichment, and the bubble size indicates the number of enriched genes

**Figure 5 ijms-20-04573-f005:**
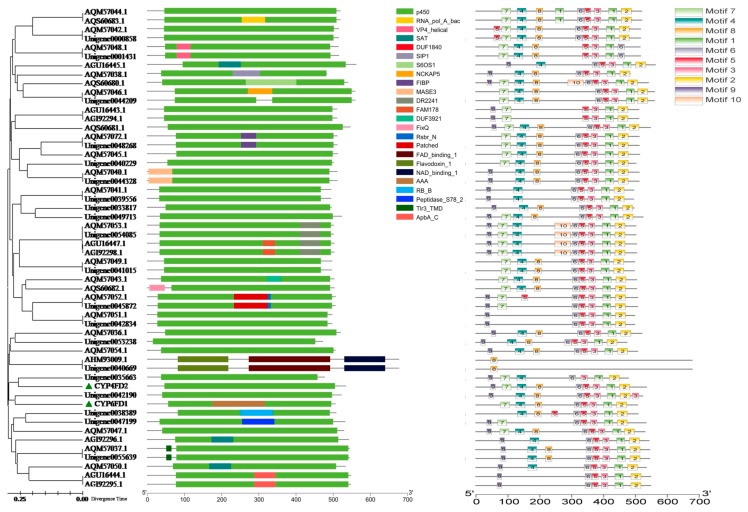
Phylogenetic tree of P450 gene family constructed by NJ method and gene molecular structure map predicted by motif search and meme.

**Figure 6 ijms-20-04573-f006:**
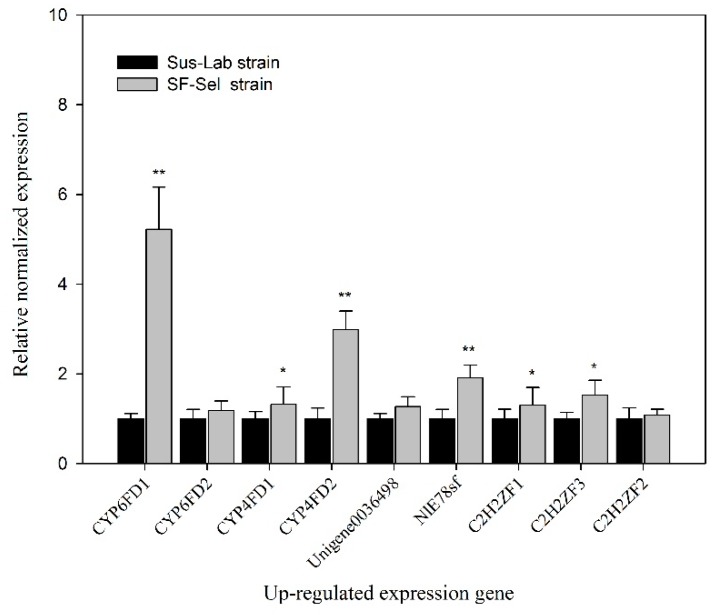
The relative expression of DEGs in Sus-Lab/SF-Sel strains. Each RT-qPCR reaction for each sample was performed in three technical replicates. Asterisks indicate significant differences of up-regulated expression gene in SF-Sel strains compared to the Sus-lab strain (Student’s *t*-test, ** *p* < 0.01 and * *p* < 0.05).

**Figure 7 ijms-20-04573-f007:**
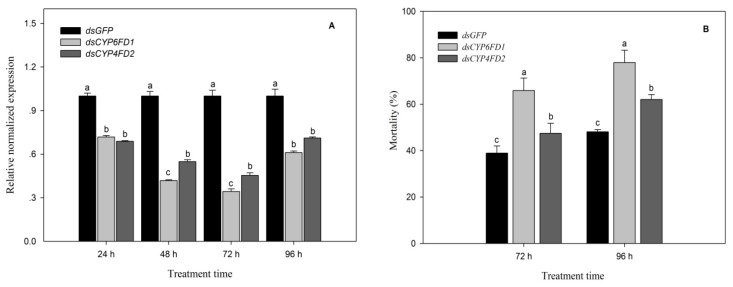
RNA interference and biological activity of two major P450 genes. (**A**)—RNA interference efficiency; (**B**)—Activity of sulfoxaflor against post-RNAi insects. Different letters (a, b, c) above bars indicate significant differences (*p* < 0.05) according to Duncan’s multiple range test. The relative normalized expression of *dsGFP*, *dsCYP6FD1* and *dsCYP4FD2* in 3^rd^-instar nymph of *S furcifera,* and the mortalities for each treatment are presented as the mean of three replications ± *SE*. Means followed by the same letters did not differ significantly (*p* > 0.05) according to the ANOVA test. The F_2, 6_ values of relative normalized expressions at 24 h, 48 h, 72 h, 96 h and mortlities at 72 h, 96 h for different treatments were 158.933, 230.377, 164.477, 51.036 and 9.949, 19.901, respectively. and the corresponding *p* values were < 0.0001, < 0.0001, < 0.0001, < 0.0001, and = 0.012 < 0.05, = 0.02 < 0.05, respectively.

**Figure 8 ijms-20-04573-f008:**
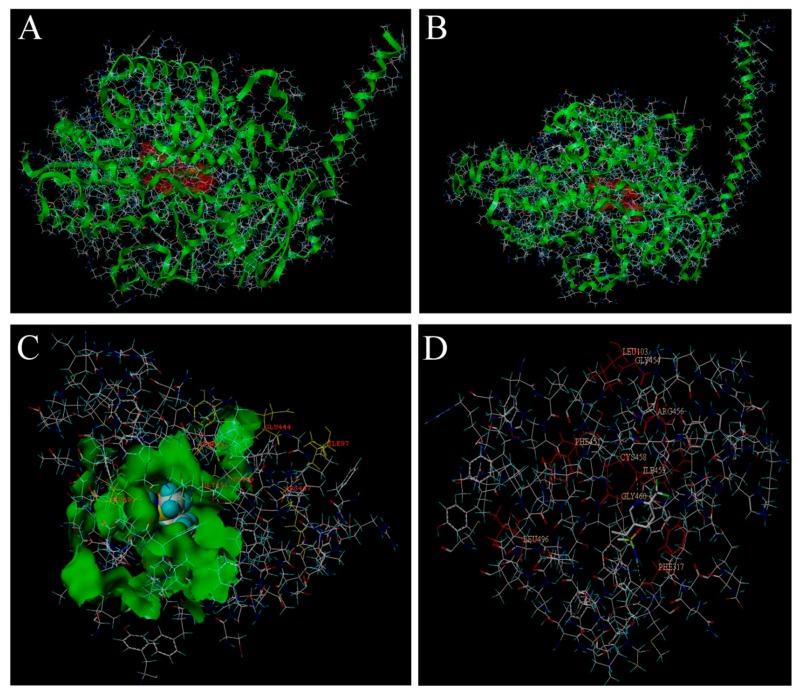
The tertiary structure of *CYP6FD1* and *CYP4FD2* and their docking structure with sulfoxaflor. (**A**) *CYP6FD1*, (**B**) *CYP4FD2*, (**C**) *CYP6FD1* domain and sulfoxaflor, (**D**) *CYP4FD2* domain and sulfoxaflor (molecular docking between sulfoxaflor and the active sites of the target P450 proteins was predicted using the Surflex-Dock program in software Syby lX-2.0 version (Tripos Inc.).

**Table 1 ijms-20-04573-t001:** Synergistic effect of three synergists and sulfoxaflor on *S. furcifera.*

Strain	Treatment	LC_50_ (µg/mL) 95%*CI*	Slope ± SE	*χ*^2^ (*df*)	SR *
Sus-Lab	sulfoxaflor	3.544(3.287–3.804)	6.499 ± 0.722	11.226(13)	/
	sulfoxaflor plus TPP	2.371(2.051–2.700)	3.253 ± 0.418	4.864(13)	1.495
	sulfoxaflor plus DEM	2.940(2.627–3.282)	4.069 ± 0.512	4.997(13)	1.205
	sulfoxaflor plus PBO	2.927(2.549–3.212)	6.893 ± 1.199	10.213(13)	1.211
SF-Sel	sulfoxaflor	7.284(6.265–8.385)	4.101 ± 0.454	14.726(13)	/
	sulfoxaflor plus TPP	2.709(2.180–3.449)	1.972 ± 0.248	4.292(13)	2.689
	sulfoxaflor plus DEM	3.191(2.529–4.061)	2.200 ± 0.307	3.268(13)	2.283
	sulfoxaflor plus PBO	1.367(1.129–1.648)	2.829 ± 0.353	4.908(13)	5.328

* SR (synergism ratio) = LC_50_ of a strain treated with sulfoxaflor alone divided by LC_50_ of the same strain treated with sulfoxaflor plus a synergist. The synergists of TPP, DEM and PBO stand for triphenyl phosphate, Diethyl maleateand piperonyl butoxide, respectively.

**Table 2 ijms-20-04573-t002:** Candidate P450 gene statistics.

GeneID	log2 Ratio(S/M)	S_vs_M Regulated	Gene Type	Annotation
*Unigene0005814*	11.17326071	UP	*CYP6FD1*	P450
*Unigene0012458*	10.38247993	UP		
*Unigene0020537*	9.861035196	UP	*CYP6FD2*	P450
*Unigene0020536*	9.83315364	UP		
*Unigene0069588*	8.681589817	UP	*CYP4FD1*	P450
*Unigene0015479*	7.934280594	UP		
*Unigene0027543*	2.07289003	UP	*CYP4FD2*	P450
*Unigene0036498*	8.17697521	UP		Transporter
*Unigene0042782*	1.136503653	UP	*C2H2ZF2*	Transcription factors
*Unigene0051504*	1.044721874	UP	*NlE78sf*	Transcription factors
*Unigene0010562*	1.411318813	UP	*C2H2ZF1*	Transcription factors
*Unigene0010210*	1.036525876	UP	*C2H2ZF3*	Transcription factors

**Table 3 ijms-20-04573-t003:** The primers of upregulation expression genes used in this study.

Gene Family	Prime	Sequence (5′-3′)	Length
Reference	*RPL9-F*	TGTGTGACCACCGAGAACAACTCA	131
	*RPL9-R*	ACGATGAGCTCGTCCTTCTGCTTT	
P450	*CYP6FD1-F*	CTTCAACATGCGGTTCACGC	187
	*CYP6FD1-R*	TTCATCCAAGCTCAACGGCT	
	*CYP4FD1-F*	AACCACTGCATGACTTTGCC	199
	*CYP4FD1-R*	TCAGCACCCGCAATGAATGT	
	*CYP6FD2-F*	GAGATGGCACACAAACCGGA	171
	*CYP6FD2-R*	GCAGAATCGCGCTAGAATGG	
	*CYP4FD2-F*	CAGCGAATGGTGGCTTCATC	183
	*CYP4FD2-R*	*ATAGCAGCCATGGTCTCACC*	
Transporter	*Unigene0036498-F*	CCCAAACCCTTCAAGACGGA	162
	*Unigene0036498-R*	GGCTGGATCGGAAATGCTCT	
Transcription factor	*NlE78sf-F*	GGAGTGTTGGGGTGGTAGTG	181
	*NlE78sf-R*	GGTGATGAACACTGCTCCGA	
	*C2H2ZF1-F*	CCATCATCAAGGCGGAACCT	182
	*C2H2ZF1-R*	ACCAGCGTTTTCAATGGTGC	
	*C2H2ZF3-F*	GTCGCCTGTGCCTTCTAGTT	165
	*C2H2ZF3-R*	AGCGGATGCACCTGATACTG	
	*C2H2ZF2-F*	ACAAGGGCATTCGCAAACAC	159
	*C2H2ZF2-R*	ATGTGCCGATCCAGATAGCG	

**Table 4 ijms-20-04573-t004:** The RNAi primers of *CYP6FD1* and *CYP4FD2* used in this study.

Prime	Sequence (5′-3′)
T7-GFP-F	TAATACGACTCACTATAGGGAAGGGCGAGGAGCTGTTCACCG
T7-GFP-R	TAATACGACTCACTATAGGGCAGCAGGACCATGTGATCGCGC
CYP6FD1dsRNAF	TAATACGACTCACTATAGGGAGAAGTCCCAATTTCACAGACGC
CYP6FD1dsRNAR	TAATACGACTCACTATAGGGAGAGATTCCGGTCTATGCGCTTC
CYP4FD2dsRNAF	TAATACGACTCACTATAGGGAGAAAGGTTTCATCTACAAAGGATTGC
CYP4FD2dsRNAR	TAATACGACTCACTATAGGGAGACATCAGTGAAATCGTGCAGAATC

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
