# Peer review of "Transcriptome Analysis of Sogatella furcifera (Homoptera: Delphacidae) in Response to Sulfoxaflor and Functional Verification of Resistance-Related P450 Genes"

_ijms, 2019, doi:10.3390/ijms20184573_

Round 1

Reviewer 1 Report

Dear Authors,

in the pdf file you can find a few comments marked in yellow.

Some major comments below

1) There were only 45 individuals per each treatments in case of mortality tests?

2) Add SD or SEM to mortality value.

3) In case of enzymatic activity test, there were only 3 biological repetition?

4) Did you check the genes expression level also in comparison to other reference genes than RPL9?

5) Check carefully the text for lacking spaces.

Kind regards

Author Response

Dear Editor and Reviewers,

   Thank for your comments on our manuscript. We have revised our manuscript according to review's suggestion. We have carefully reviewed all comments and suggestions and answered those questions point-by-point and expected the revised manuscript was acceptable for publication. The whole changes have been modified with model revision and marked with GREEN in our paper. We look forward to your positive response.

Yours sincerely,

Dr. Wang

Reviewer 1

Comment #1: Dear Authors, in the pdf file you can find a few comments marked in yellow.

Answer: We have revised the WHOLE manuscript carefully and tried to avoid any grammar or syntax error or add “space” before some words.

Comment #2: There were only 45 individuals per each treatment in case of mortality tests?

Answer: Yes. When we assayed the toxities of sulfoxaflor to S. turcifera, five to seven centrations and control were preformed, and 45 individuals were pepared for per each treatment. However, when we checked the biological function of CYP6FD1 and CYP4FD2 via RNA interference, sixty third-instar nymphs were used for each treatment.

Comment #3: Add SD or SEM to mortality value.

Answer: Yes. We have added the SD or SEM to mortality value.

Comment #4: In case of enzymatic activity test, there were only 3 biological repetitions?

Answer: Yes. We have performed 3 biological repetitions for each treatment.

Comment #5: Did you check the genes expression level also in comparison to other reference genes than RPL9?

Answer: No. When we search the reference genes, we searched several papers which evaluated the stability of common internal reference genes in different tissues of Sogatella furcifera, Laodelphax striatellus, and Nilaparvata lugens, such as “He, X.T.; Liu, C.C.; Li, Z.Q.; et al. Validation of reference genes for quantitative real-time PCR in Laodelphax striatellus. J.Integr. Agr. 2014, 13, 811-818.”, “Miao, Y.T.; Jia, H.K.; Li, Z.; et al. Transcriptomic and expression analysis of the salivary glands in brown planthoppers, Nilaparvata lugens (Hemiptera: Delphacidae). J. Econ.Entomol. 2018, 111, 2884-2893” and “An, X.K.; Hou, M.L.; Liu, Y. D. Reference gene selection and evaluation for gene expression studies using qRT-PCR in the white-backed planthopper, Sogatella furcifera (Hemiptera: Delphacidae). J. Econ. Entomol. 2016, 109, 879”. All results indicated that RPL9 had a higher stability in different tissues. Meanwhile, our transcriptome data (following table) also showed that RPL9 (Symbol) had a higher stability (FDR=1) and expression than other reference genes in tested M and S strains. Therefore, we selected RpL9 as our housekeep gene. Maybe we should set two or three reference genes in the future

GeneID

M1_rpkm

M2_rpkm

M3_rpkm

S1_rpkm

S2_rpkm

S3_rpkm

log2 Ratio(S/M)

P-value

FDR

Symbol

Unigene0040855

722.9935

801.7463

693.9832

757.8963

727.8345

632.4495

-0.0669

0.657205

1.00

RpL9

Unigene0033687

1955.134

2156.405

1809.041

1918.351

1782.827

1602.352

-0.15879

0.295029

0.902409

RpS18

Unigene0007839

8.4336

8.4507

9.6405

6.1554

6.3704

7.1893

-0.42804

0.024328

0.369868

Gapdh

Unigene0047920

246.1355

219.521

285.066

94.6263

126.092

110.0663

-1.18239

5.47E-12

5.37E-09

actin 2

Unigene0040279

0.5923

0.3568

0.5063

0.3764

4.5547

0.2929

1.843739

0.041497

0.475014

Actin

Comment #6: Check carefully the text for lacking spaces.

Answer: Agree. We have checked all the lacking speaces in the manuscript.

Reviewer 2 Report

This paper by Wang et al. describes the development of a lab strain of S. turcifera resistant to the insecticide sulfoxaflor, and the authors begin to elucidate the mechanisms responsible for the insecticide resistance using transcriptome analysis and confirming by RT-PCR and RNAi analyses, as well as examining the tertiary structure and binding site of the key upregulated  P450s that were identified.

Overall the paper has some important and interesting findings and the analysis is thorough. However, there are too many figures, and many do not present data that significantly adds to the results section, and therefore authors should consider moving some of these to supplementary materials to increase the impact and readability of the paper.

The introduction is good but is missing some key information on detoxification enzymes, which are an important theme of this paper and represent a significant part of the results.

The Discussion is generally good, but would benefit from a concluding statement to highlight the key findings and put them in perspective of the bigger picture of development of insecticide resistance.

Some tables are missing from Materials and methods, and some of the Results and figures seem to have some errors.

Abstract:

Line 10: include the aphid’s common name.

Line 24: replace “meanwhile in which of” with “and mortality in” so that the sentence is more easily understood.

Introduction:

The introduction is overall good, but is missing some general information on detoxification enzymes and why these are important in resistance to insecticides.

Results:

Line 84: It would be useful to briefly introduce the synergistic agents and which enzyme they are acting on or their mechanism of action.

Line 86 & 89: both lines state that the toxicity decreased with synergist treatments, but toxicity increased since LC50 is reduced.

Line 96: please give a brief introduction to what CarEs, GSTS and P450s are, as these have not been previously discussed so it is not clear why these are being tested. Ideally add some information on these to the Introduction section, as mentioned above.

Line 98, 98, 106, 107: the HN strain is mentioned although this hasn’t been mentioned previously – presumably these should be corrected to Sus-Lab strain.

Line 97-99: this sentence states “the activity of CarEs in the SF-Sel strain treated with TPP was the highest(0.9616 mmol/min mg pro), followed by those in the HN strain (0.8257 mmol.min-1.mg pro-1), the SF-Sel strain (0.8187 mmol.min-1.mg pro-1) and the HN strain (0.7397 mmol.min-1.mg pro-1) treated with TPP.”

But the order is different from what is shown on Figure 1A where SF-Sel with TPP is highest, then Sus-Lab with TPP, then SF-Sel, then Sus-Lab.

Line 100-103: This sentence says CarE activity, but is describing GST activity – please correct CarE to GST.

Figure 1: x-axis labels are unclear as the text is so close together. It would be good to clarify these by putting the labels at a 45degree angle to the x axis, for example.

Line 111: the spacing in the numbers is confusing. They should be written 335,119,142 and 50,653,904. Also on line 115 the large number should be expressed as 329,449,482.

Table 2: the authors should consider moving this table to supplementary materials. I don’t believe the data presented here are important or interesting enough for the main results section.

Figure 2: Again this is possibility not a useful figure for the main results, and could be put in supplementary figures.

Figure 3: All of Figure 3 could be supplementary figures, the data are not really significant enough for the main results section.

Figure 4 and 5: These could be combined into a single figure with the heat map part A and the PCA as part B.

Figure 5: the key on the right-hand side of this figure is not necessary because the dots in the PCA are individually labelled.

Figure 6: Number of up-regulated genes in the bar chart is 557 but in the text (line 145) is 559. Which is correct?

Also the results in the bar chart don’t seem to match the heat map. In the heat map there are more genes up-regulated in S strains, so perhaps the bar chart should be labelled S-vs-M (as in table 3) and not M-vs-S?

Line 150: Lab-HN strain – is this Sus-Lab?

Figure 7: Consider moving this to supplementary figures as well.
As there are more up- than down-regulated genes on the chart should this be S-vs-M not M-vs-S? Also the labels on the x-axis showing the GO terms are too small to read unless the reader zooms right in.

Table 3: It could be made clearer in the table which genes are P450s and transcription factors, perhaps by adding an additional column titled “annotation”.

Line 173: What is AMH93009.1? This has not been mentioned before. Also in the figure it is called AHM93009.1.

Line 182: As CYP6FD1 and CYP4FD2 are labelled by their unigene numbers in Figure 9, it is really difficult to find them in the phylogenetic tree. It would be useful if the authors included their unigene numbers in brackets after each mention in line 182 i.e. CYP6FD1 (Unigene0005814) and also highlighted these 2 unigenes in the figure, e.g. by bold type or with an asterisk or symbol.

Line 182: “domains of motif9 and AAA” – what is AAA? Should it say motif 10?

Figure 10: I suggest that this figure also be Supplementary material.

Line 207: check the title, this section doesn’t seem to describe sequence analysis, but rather analysis of tertiary structure.

Figure 13: The key areas and amino acids of the structure could be better labelled in C and D, as they are difficult to see without zooming into the image.

Discussion

Line 262: check spelling – mortality

It would be nice to have some short conclusions (a couple of sentences) summarising the main findings of the paper and putting them in context of the bigger picture of the development of sulfoxaflor resistance in S. furcifera and how this might be overcome. For example, the main findings show that it is likely that S. furcifera will develop resistance to sulfoxaflor and that upregulation of detoxification enzymes such as P450s is a likely mechanism. However, the authors also show that the toxicity of sulfoxaflor is increased by using synergistic agents, so perhaps this is one possible approach that could be used in the field to prevent rapid development of resistance to this compound.

Materials & Methods

Line 309: “many nymphs” - Please be more specific, stating the number of nymphs or if not exact, a guideline amount such as 10-20 nymphs or approximately 100 nymphs.

Line 358: HN-Sel and HN-Lab are mentioned. Are these different strains or are they actually SF-Sel and Sus-Lab?

Line 389: Table 4 is missing from the manuscript.

Line 399: Table 5 is missing from the manuscript.

Line 414: check the title, this section doesn’t seem to describe sequence analysis.

Author Response

Dear Editor and Reviewers,

   Thank for your comments on our manuscript. We have revised our manuscript according to review's suggestion. We have carefully reviewed all comments and suggestions and answered those questions point-by-point and expected the revised manuscript was acceptable for publication. The whole changes have been modified with model revision and marked with GREEN in our paper. We look forward to your positive response.

Yours sincerely,

Dr. Wang

Reviewer 2#

Comment #1: This paper by Wang et al. describes the development of a lab strain of S. turcifera resistant to the insecticide sulfoxaflor, and the authors begin to elucidate the mechanisms responsible for the insecticide resistance using transcriptome analysis and confirming by RT-PCR and RNAi analyses, as well as examining the tertiary structure and binding site of the key upregulated P450s that were identified. Overall the paper has some important and interesting findings and the analysis is thorough. However, there are too many figures, and many do not present data that significantly adds to the results section, and therefore authors should consider moving some of these to supplementary materials to increase the impact and readability of the paper.The introduction is good but is missing some key information on detoxification enzymes, which are an important theme of this paper and represent a significant part of the results.

The Discussion is generally good, but would benefit from a concluding statement to highlight the key findings and put them in perspective of the bigger picture of development of insecticide resistance. Some tables are missing from Materials and methods, and some of the Results and figures seem to have some errors.

Answer: Agree. We have revised the WHOLE manuscript carefully and tried to avoid any grammar or syntax error. We also add some key information on detoxification enzymes in the introduction part. Tables 3 and 4 have been added and some of the Results and figures have been carefully checked or redrawn in our newer manuscript.

Comment #2: Line 10: include the aphid’s common name.

Answer: Agree. We have added the common name of “The white-back planthopper (WBPH), Sogatella furcifera”. We also added some highlight the key findings and put them in perspective of the bigger picture of development of insecticide resistance. Some tables are missing from Materials and methods, and some of the Results and figures seem to have some errors.

Comment #3: Line 24: replace “meanwhile in which of” with “and mortality in” so that the sentence is more easily understood.

Answer: Agree. We have changed the “meanwhile in which of” to “and mortality in”.

Comment #4: Introduction: The introduction is overall good, but is missing some general information on detoxification enzymes and why these are important in resistance to insecticides.

Answer: Agree. We have added the information on detoxification enzymes.

Comment #5: Results: Line 84: It would be useful to briefly introduce the synergistic agents and which enzyme they are acting on or their mechanism of action.

Answer: Agree. We have added which enzyme of synergistic agent acting on.

Comment #6: Line 86 & 89: both lines state that the toxicity decreased with synergist treatments, but toxicity increased since LC50 is reduced.

Answer: We have a wrong expression and the “decreased” has been changed to “enhanced”.

 Comment #7: Line 96: please give a brief introduction to what CarEs, GSTS and P450s are, as these have not been previously discussed so it is not clear why these are being tested. Ideally add some information on these to the Introduction section, as mentioned above.

   Answer: Agree. We have added a brief introduction on the CarEs, GSTS and P450s.

Comment #8: Line 98, 98, 106, 107: the HN strain is mentioned although this hasn’t been mentioned previously – presumably these should be corrected to Sus-Lab strain.

   Answer: Agree. We have changed all “HN strain” to “Sus-Lab strain”.

Comment #9: Line 97-99: this sentence states “the activity of CarEs in the SF-Sel strain treated with TPP was the highest(0.9616 mmol/min mg pro), followed by those in the HN strain (0.8257 mmol.min-1.mg pro-1), the SF-Sel strain (0.8187 mmol.min-1.mg pro-1) and the HN strain (0.7397 mmol.min-1.mg pro-1) treated with TPP.” But the order is different from what is shown on Figure 1A where SF-Sel with TPP is highest, then Sus-Lab with TPP, then SF-Sel, then Sus-Lab.

       Answer: We have made misorder and checked the order with SF-Sel with TPP is highest, then SF-Sel, then Sus-Lab with TPP, then Sus-Lab.

Comment #10: Line 100-103: This sentence says CarE activity, but is describing GST activity. Please correct CarE to GST.

Answer: Agree. We have changed “CarEs” to “GSTs”.

Comment #11: Figure 1: x-axis labels are unclear as the text is so close together. It would be good to clarify these by putting the labels at a 45degree angle to the x axis, for example.

Answer: Agree. We have put the labels at a 45 degree angle to the x axis to make the x-axis label clearer.

Comment #12: Line 111: the spacing in the numbers is confusing. They should be written 335,119,142 and 50,653,904. Also on line 115 the large number should be expressed as 329,449,482.

Answer: Agree. We have rewritten those numbers.

Comment #13: Table 2: the authors should consider moving this table to supplementary materials. I don’t believe the data presented here are important or interesting enough for the main results section.

Answer: Agree. We have move Table 2 to the Supplementary Table 1.

Comment #14: Figure 2: Again this is possibility not a useful figure for the main results, and could be put in supplementary figures.

Answer: Agree. We have move Figure 2 to the Supplementary Figures 1.

Comment #15: Figure 3: All of Figure 3 could be Supplementary figure; the data are not really significant enough for the main results section.

Answer: Agree. We have move Figure 2 to the Supplementary Figures 2.

Comment #16: Figure 4 and 5: These could be combined into a single figure with the heat map part A and the PCA as part B. Figure 5: the key on the right-hand side of this figure is not necessary because the dots in the PCA are individually labelled.

Answer: Agree. We have combined Figure 4 and 5 into a single figure, and deleted the key on the right-hand side of this figure.

Comment #17: Figure 6: Number of up-regulated genes in the bar chart is 557 but in the text (line 145) is 559. Which is correct?

Answer: We have changed “559” to “557”.

Comment #18: Also the results in the bar chart don’t seem to match the heat map. In the heat map there are more genes up-regulated in S strains, so perhaps the bar chart should be labelled S-vs-M (as in table 3) and not M-vs-S?

Answer: Agree. We have changed “M-vs-S” to “S-vs-M”.

Comment #19: Line 150: Lab-HN strain – is this Sus-Lab?

Answer: We have changed “Lab-HN strain” to “Sus-Lab strain”.

Comment #20: Figure 7: Consider moving this to supplementary figures as well.
As there are more up- than down-regulated genes on the chart should this be S-vs-M not M-vs-S? Also the labels on the x-axis showing the GO terms are too small to read unless the reader zooms right in.

       Answer: Agree. We have move Figure 2 to the supplementary figures 3, and have changed “M-vs-S” to “S-vs-M”.

Comment #21: Table 3: It could be made clearer in the table which genes are P450s and transcription factors, perhaps by adding an additional column titled “annotation”.

       Answer: Agree. We have added an additional column titled “annotation” in Table 3 to explain which genes are P450s and transcription factors.

Comment #22: Line 173: What is AMH93009.1? This has not been mentioned before. Also in the figure it is called AHM93009.1.

Answer: We had a spelling mistake and the “AMH93009.1” has been changed to “AHM93009.1”, which is the sequences of NADPH-cytochrome P450 reductase in Sogatella furcifera downloaded from NCBI.

Comment #23: Line 182: As CYP6FD1 and CYP4FD2 are labelled by their unigene numbers in Figure 9, it is really difficult to find them in the phylogenetic tree. It would be useful if the authors included their unigene numbers in brackets after each mention in line 182 i.e. CYP6FD1 (Unigene0005814) and also highlighted these 2 unigenes in the figure, e.g. by bold type or with an asterisk or symbol.

Answer: We have redrawn the Figure9 to ensure it clearer.

Comment #24: Line 182: “domains of motif9 and AAA” – what is AAA? Should it say motif 10?

Answer: No. Ten motifs (motif 1~motif 10, respectively) composed of very conservative amino acid residues were found in 54 of P450 genes of white-backed planthopper by meme search, but the motifs annotated as AAA was found by motif search, which contains several distinct features in addition to the conserved alpha-beta-alpha core domain structure and the Walker A and B motifs of the P-loop NTPases.

Comment #25: Figure 10: I suggest that this figure also be Supplementary material.

Answer: Agree. We have move Figure 10 to the supplementary figures 4.

Comment #26: Line 207: check the title, this section doesn’t seem to describe sequence analysis, but rather analysis of tertiary structure.

Answer: Agree. We have changed the title to “Interaction of tertiary structure of CYP6FD1 and CYP4FD2 with sulfoxaflor”

Comment #27: Figure 13: The key areas and amino acids of the structure could be better labelled in C and D, as they are difficult to see without zooming into the image.

Answer: Agree. We have labeled the structures of amino acids in key areas of Figure C and D

 Discussion

Comment #28: Line 262: check spelling – mortality

Answer: We have changed the word “morality” to “mortality”.

Comment #29: It would be nice to have some short conclusions (a couple of sentences) summarising the main findings of the paper and putting them in context of the bigger picture of the development of sulfoxaflor resistance in S. furcifera and how this might be overcome. For example, the main findings show that it is likely that S. furcifera will develop resistance to sulfoxaflor and that upregulation of detoxification enzymes such as P450s is a likely mechanism. However, the authors also show that the toxicity of sulfoxaflor is increased by using synergistic agents, so perhaps this is one possible approach that could be used in the field to prevent rapid development of resistance to this compound.

Answer: Agree. We have added some short conclusions to summary the main findings of the paper and which should be further study in the future.

Materials & Methods

Comment #30: Line 309: “many nymphs” - Please be more specific, stating the number of nymphs or if not exact, a guideline amount such as 10-20 nymphs or approximately 100 nymphs.

Answer: Approximately 300 of 3rd-instar nymphs were transferred to the rice seedlings treated with each synergistic agent for approximately 2 h.

Comment #31: Line 358: HN-Sel and HN-Lab are mentioned. Are these different strains or are they actually SF-Sel and Sus-Lab?

Answer: We have changed the HN-Sel and HN-Lab strain to SF-Sel and Sus-Lab strains respectively.

Comment #32: Line 389: Table 4 is missing from the manuscript.

Answer: We have added the Table 4 (in fact Table 3) in the newer manuscript.

Comment #33: Line 399: Table 5 is missing from the manuscript.

Answer: We have added the Table 5 (in fact Table 4) in the newer manuscript.

Comment #34: Line 414: check the title, this section doesn’t seem to describe sequence analysis.

Answer: Agree. We have adjusted the title name to “Prediction the interaction of tertiary structure of CYP6FD1 and CYP4FD2 with sulfoxaflor”

Reviewer 3 Report

Authors present an extensive transcriptome analysis on a major Chinese rice pest, Sogatella furcifery, to understand its resistance against a new diamide insecticide, Sulfoxaflor, and could evaluate the overexpression of two P450 genes as primary factors in resistance development. The experiments are carefully done and the results are of general interest. A main concern, however, is that the manuscript is rather difficult to read, at least in parts, and that some of the figures are not easy to understand because of the lack of detailed figure legends. Some examples:

lines 24/25: the sentence is unclear

line 84: acronyms should be explained, when first used

GST activity in Fig. 1B is not mentioned in the text

lines 148/149: the sentence is unclear

line 256: sentence unclear

line 289 and others: authors should use uniform units and not e.g., mg.L-1 or mg/l or mM and mol/L etc.

line 305: explain names of synergistic agents

line 308: sentence incomplete

line 309: give number of 3rd instars used

line 319: centrifugation steps have to be given in g values, not rpm

lines 364-367: explanation for RPKM is not clear

line 403-405: when the concentration of the injected solution is 4 ng/uL and 120 ng were injected, then animals have received 30 uL but not 30 nL

Table 1: names of synergists have to be explained in the figure legend

Legend to Figure 1:  explain statistics

Legend to Figure 2: explain A and B

Legend to Figure 6: incomplete sentence

Legend to Figure 7: ordinate is the Y-axis where number of genes are shown

Legend to Figure 10 is not clear. What does 1 to 10 mean?  

Minors:

The manuscript has to be carefully checked for spaces between words, numbers and units etc.

line 113: is 1.35% the correct value?

line 145: is 557 the correct value?

line 210: where heme binding occurs

References have to be checked for abbreviation of the journal titles. Species names have to be given in italics.

Author Response

Dear Editor and Reviewers,

   Thank for your comments on our manuscript. We have revised our manuscript according to review's suggestion. We have carefully reviewed all comments and suggestions and answered those questions point-by-point and expected the revised manuscript was acceptable for publication. The whole changes have been modified with model revision and marked with GREEN in our paper. We look forward to your positive response.

Yours sincerely,

Dr. Wang

Reviewer 3#

Authors present an extensive transcriptome analysis on a major Chinese rice pest, Sogatella furcifery, to understand its resistance against a new diamide insecticide, Sulfoxaflor, and could evaluate the overexpression of two P450 genes as primary factors in resistance development. The experiments are carefully done and the results are of general interest. A main concern, however, is that the manuscript is rather difficult to read, at least in parts, and that some of the figures are not easy to understand because of the lack of detailed figure legends. Some examples:

Comment #1: lines 24/25: the sentence is unclear

Answer: Agree. We have changed “meanwhile in which of” to “and mortality in”.

Comment #2: line 84: acronyms should be explained, when first used

Answer: all acronyms have been explained when first used.

Comment #3: GST activity in Fig. 1B is not mentioned in the text

Answer: We have miswritten the GSTs to CarEs and from line 100-103 we described the GST activity.

Comment #4: lines 148/149: the sentence is unclear

Answer: The sentence has been changed to “The up- or downregulated DEGs were enriched and assessed in the three Gene Ontology (GO) categories of biological process, cellular component and molecular function”.

Comment #5: line 256: sentence unclear

Answer: The sentence has been changed to “the reducing expression of CYP6ER1 in sulfoxaflor-resistant strain through RNAi could significantly increase its’ susceptibility to sulfoxaflor”.

Comment #6: line 289 and others: authors should use uniform units and not e.g., mg.L-1 or mg/l or mM and mol/L etc.

Answer: We have checked the whole manuscript and ensured use uniform units.

Comment #7: line 305: explain names of synergistic agents

Answer: We have explained the names of synergistic agents firstly use in the part of Synergism Experiment of our newer manuscript, including triphenyl phosphate (TPP), Diethyl maleate (DEM) and piperonyl butoxide (PBO).

Comment #8: line 308: sentence incomplete

Answer: We have changed the sentence to “after which rice seedlings into the synergistic treatment solutions for 30 s and naturally dried them.”

Comment #9: line 309: give number of 3rd instars used

Answer: approximately 300 of 3rd-instar nymphs were prepared for each synergistic agent.

Comment #10: line 319: centrifugation steps have to be given in g values, not rpm

Answer: We have performed the centrifugation at 4 °C, 10,000 × g for 15 min using a 5417R centrifuge (Eppendorf, Germany).

Comment #11: lines 364-367: explanation for RPKM is not clear

Answer: The RPKM(A) value stands for the expression of gene A; C values stands for the number of reads that uniquely aligned to gene A; N values stands for the total number of reads that uniquely aligned to all genes, and L stands for the number of bases on gene A. The RPKM method is able to eliminate the influence of different gene length and sequencing data amount on the calculation of gene expression. Therefore, the calculated gene expression can be directly used for comparing the difference of gene expression among samples.

Comment #12: line 403-405: when the concentration of the injected solution is 4 ng/uL and 120 ng were injected, then animals have received 30 uL but not 30 nL

Answer: Agree. We have changed the unit nl to μl.

Comment #13:Table 1: names of synergists have to be explained in the figure legend

Answer: Agree. The synergists of TPP, DEM and PBO stand for triphenyl phosphate, Diethyl maleateand piperonyl butoxide, respectively.

Comment #14: Legend to Figure 1:  explain statistics

Answer: Agree. The activities of CarE, GST and P450 in 3rd-instar nymph of S furcifera are presented as the mean of three replications ± SE. Means followed by the same letters did not differ significantly (p > 0.05) according to the ANOVA test. The F3, 8 values of different treatments on CarE, GST and P450 in 3rd-instar nymph of S furcifera were 1.818, 5.410, 610.745, and the P values on CarE, GST and P450 in 3rd-instar nymph of S furcifera were =0.222 > 0.05, =0.025< 0.05, =< 0.0001, respectively.

Comment #15: Legend to Figure 2: explain A and B

Answer: Agree. (A) Distribution of unigenes in different length ranges (e.g.,<200 represents 0–200). (B) Distribution of unigenes with different number of reads (e.g.,1-10 represents unigene consisted with 1-10 reads).

Comment #16: Legend to Figure 6: incomplete sentence

      Answer: We have changed the sentence to “The statistical and cluster heat maps of DEGs among M-vs-S.” and “Color-scaled represents log 2 (fold change) values for resistant lines’’.

Comment #17: Legend to Figure 7: ordinate is the Y-axis where number of genes are shown

Answer: Agree.We had a spelling mistake and the “The ordinate means GO terms” has been changed to “The abscissa means GO terms”, while the “the abscissa means the number of DEGs of each GO term” has been changed to “the ordinate means the number of DEGs of each GO term”.

Comment #18: Legend to Figure 10 is not clear. What does 1 to 10 mean?

Answer: Figure 10 (Supplementary Figures 4) was conservative sequence display of different motif structures searched by the meme. 1 to 10 mean the identifier of different motif structures, and abscissa represents conserved amino acid sequences of different motifs

Minors:

Comment #1: The manuscript has to be carefully checked for spaces between words, numbers and units etc.

      Answer: we have carefully checked for spaces between words, numbers and units etc.

Comment #2: line 113: is 1.35% the correct value?

      Answer: Yes. We checked the data in Supplementary Table 1 and the proportion of low-quality reads was 1.23%~1.35%.

Comment #3: line 145: is 557 the correct value?

      Answer: We have checked the data and there were 557 of upregulated DEGs in the SF-Sel strain.

Comment #4: line 210: where heme binding occurs

      Answer: In the structure of CYP6FD1, the absolutely conserved cysteines (FGEGPRYCIG) of the heme-binding loop were also located near the entrance of the active pocket (amino acid residues labeled Red in Figure A). In the structure of CYP4FD2, the absolutely conserved cysteines (FSAGPRNCIG) of the heme-binding loop were also located near the entrance of the active pocket (amino acid residues labeled Red in Figure B).

Figure A. The tertiary structure of CYP6FD1.

Figure B. The tertiary structure of CYP4FD2.

Comment #5: References have to be checked for abbreviation of the journal titles. Species names have to be given in italics.

      Answer: yes. We have carefully checked the references.
